# Feature-Weighted Sampling for Proper Evaluation of Classification Models

Hyunseok Shin [1] and Sejong Oh [2,*]

1   Department of Computer Science, Dankook University 152, Yongin 16890, Korea; shinhseok@gmail.com
2   Department of Software Science, Dankook University, Yongin 16890, Korea
*   Correspondence: sejongoh@dankook.ac.kr; Tel.: +82-31-8005-3222

**Abstract:** In machine learning applications, classification schemes have been widely used for prediction tasks. Typically, to develop a prediction model, the given dataset is divided into training and test sets; the training set is used to build the model and the test set is used to evaluate the model. Furthermore, random sampling is traditionally used to divide datasets. The problem, however, is that the performance of the model is evaluated differently depending on how we divide the training and test sets. Therefore, in this study, we proposed an improved sampling method for the accurate evaluation of a classification model. We first generated numerous candidate cases of train/test sets using the R-value-based sampling method. We evaluated the similarity of distributions of the candidate cases with the whole dataset, and the case with the smallest distribution–difference was selected as the final train/test set. Histograms and feature importance were used to evaluate the similarity of distributions. The proposed method produces more proper training and test sets than previous sampling methods, including random and non-random sampling.

**Keywords:** classification; training and test sets; sampling; feature importance; evaluation

## 1. Introduction

Classification problems in machine learning can be easily found in the real world. Doctors diagnose patients as either diseased or healthy based on the symptoms of a specific disease in the past, and in online commerce, security experts decide whether transactions are fraudulent or normal based on the pattern of previous transactions. As in this example, the purpose of classification in machine learning is to predict unknown features based on past data. An explicit classification target, such as "diseased" or "healthy", is called a class label. The classification belongs to supervised learning because it uses a class label. Representative classification algorithms include decision trees, artificial neural networks (ANNs), naive Bayes (NB) classifiers, support vector machine (SVM), and k-nearest neighbors (KNN) [1].

In general, the development of a classification model comprises two phases, as shown in Figure 1, starting with data partitioning. The entire dataset is divided into a training set and a test set, each of which is used during different stages and for different purposes. The first is the learning or training phase using the training set. At this time, part of the training set is used as a validation set. The second phase is the model evaluation phase using the test set. The evaluation result using a test set is considered the final performance of the trained model. The inherent problem in the development of a classification model is that the model's performance (accuracy) inevitably depends on the divided training and test set. This is because the model reflects the characteristics of the training set, but the accuracy of the model is influenced by the characteristics of the test set. If a model with poor actual performance is evaluated with an easy-to-classify test set, the model performance will look good. Conversely, if a model with good performance is evaluated by a difficult-to-classify test set, the model performance will be underestimated. In our previous work [2], we showed that 1000 cases of train/test sets by random sampling

produced different classification accuracies from 0.848 to 0.975. This phenomenon is due to the difference in data distribution between the training and test sets, emphasizing that dividing the entire dataset into training and test sets has a significant impact on model performance evaluation.

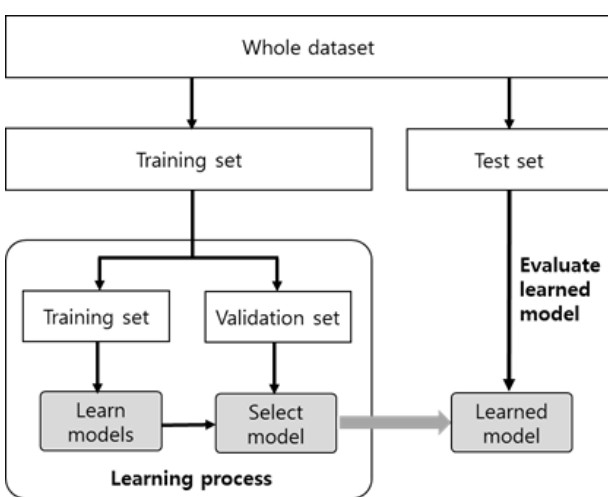

**Figure 1.** Development process of a classification model.

The ideal goal of splitting train/test sets is that the distributions of both the training and test sets become the same as the whole dataset. However, this is a difficult task for multi-dimensional datasets. Various methods have been proposed to solve this problem. Random sampling is an easy and widely used method. In random sampling, each data instance has the same probability of being chosen, and this can reduce the bias of model performance. However, it produces a high variance in model performance, if a dataset has an abnormal distribution or the size of the sample is small [3,4]. Systematic sampling is a method of extracting data by randomly arranging data and skipping at regular intervals [5]. Stratified sampling is a method of first dividing a population into layers, so that they do not overlap, and then sampling from each layer. It uses the internal structure (layers) and the distribution of a dataset [4]. D-optimal [6] and the most descriptive compound method (MDC) [7] are advanced stratified sampling methods. The potential error of the descriptor and rank sum of the distance between compounds are the internal structures of D-optimal and MDC, respectively.

R-value-based sampling (RBS) [2] is a type of stratified sampling. It divides the entire dataset into $n$ groups (layers) according to the ratio of "class overlap", and applies systematic sampling to each group. In general, the classification accuracy for a dataset is strongly influenced by the degree of overlap of the classes in the dataset [4,8]. The degree of class overlap was measured using the R-value [8]. Let us suppose a data instance $p$ and $q_1, q_2, \ldots, q_k$ are the $k$-nearest neighbor instances of $p$. If $r$ is the number of instances that belong to the $k$-nearest neighbors and their class labels are different from that of $p$, the degree of overlap of $p$ is $r$ $(0 \leq r \leq k)$. In other words, $p$ belongs to group $r$. The experimental results confirm that RBS produces better training and test sets than random and several non-random sampling methods.

In the machine learning area, $k$-fold cross-validation has been used to overcome the overfitting problem in classification. It makes $k$ training models, and the mean of test accuracies is considered as an evaluation measure for parameter tuning of a model or comparison of different models. The repeated holdout method, also known as Monte Carlo cross-validation, is also available for model evaluation [3,9]. During the iteration of the holdout process, the dataset is randomly divided into training and test sets, and the mean of the model accuracy gradually converges to one value [2]. The purpose of k-fold cross-validation and the holdout method is different from that of the sampling methods. Both k-fold cross-validation and holdout methods produce multiple train/test sets, and

as a result, they make multiple prediction models. We cannot know which is a desirable model. Therefore, they were excluded from the discussion of the sampling issue.

In this study, we propose an improved sampling method based on RBS. We generated candidate train/test sets using the modified RBS algorithm and evaluated the distribution similarity between the candidates and the whole dataset. In the evaluation process, a data histogram and feature importance were considered. Finally, the case with the smallest deviation of the distribution was selected. We compared the proposed method with RBS, and we confirmed that the proposed method shows better performance than the previous RBS.

## 2. Materials and Methods

As mentioned earlier, the ideal training and test sets should have the same distribution as the original dataset. To achieve this goal, we propose a method called feature-weighted sampling (FWS). Our main idea is as follows:

(1)    Generate numerous candidate cases of train/test sets using modified RBS.
(2)    Evaluate the similarity between the original dataset and candidate cases. The similarity is measured by the distance.
(3)    Choose the case that has smallest distance to original dataset.

Figure 2 summarizes the proposed method in detail. The first phase generates $n$ train/test set candidates with stratified random sampling. Stratified sampling uses the modified RBS method, which reflects the amorphic property of the data, called class overlap. The second step is to select one of the candidates with the distribution that is most similar to the original dataset. To evaluate the similarity of distribution, we measured the distance between the train/test sets and the original dataset in terms of distance. To calculate the distance between the original dataset and train/test sets, we tested Bhattacharyya distance [10], histogram intersection [11], and Earth Mover's Distance [12]. Finally, we adopted the Earth Mover's Distance. Feature importance was applied to the weighting feature during the distance calculation. As a result, the train/test sets that had the smallest distance from the original dataset were selected. For the evaluation of the sampling method, we devised a metric named the mean accuracy index (*MAI*). Using the *MAI*, we compared the proposed FWS and RBS. Twenty benchmark datasets and four classifiers, including *k*-nearest neighbor (KNN), support vector machine (SVM), random forest (RF), and C50 were used for the comparison.

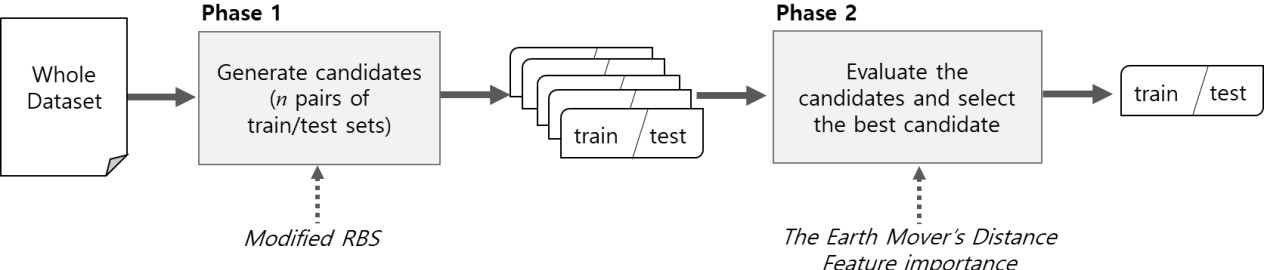

**Figure 2.** Overall process of the proposed sampling method.

### 2.1. Phase 1: Generate Candidates

In the candidate generation step, 1000 candidates with a pair of train/test sets were generated using a modified RBS. Next, 25% and 75% of the total instances were sampled to the test set and training set, respectively. Class overlap is the key concept of an RBS. We first summarize the class overlap and explain the modified RBS.

### 2.1.1. Concept of Class Overlap

Class overlap refers to the overlap of data instances among classes, and wide class overlap makes it difficult to classify tasks [8]. The overlap number of an instance $p$ is calculated by counting the number of instances with different class labels in the $k$-nearest neighbors. Figure 3 shows the class overlap value for a data instance (red cross in Figure 3) when $k = 3$. If the overlap number is over the threshold, we can determine that $p$ is located in the overlapped area. The ratio of instances located in the overlapped area is the R-value [8]. The R-value can be used to evaluate the quality of the datasets. In RBS, the overlap number of an instance is used to group the instance. If $k = 3$, then an instance can belong to one of the four groups. The RBS performs sampling train/test instances from the four groups.

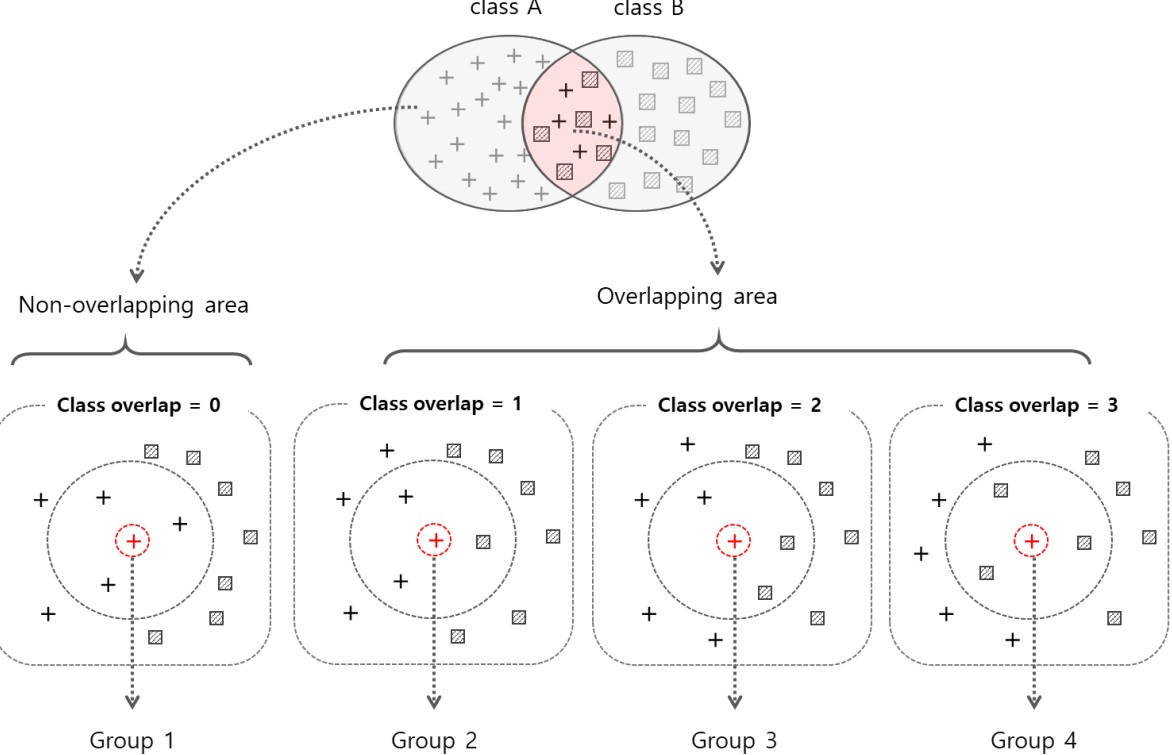

**Figure 3.** An example of grouping of data instances according to the class overlap number where $k = 3$.

### 2.1.2. Modified RBS

The original RBS adopts a stratified sampling method. It groups each instance according to the class overlap number and then samples each group in a stratified manner. As a result, the original RBS always produces the same training and test sets. We replaced the stratified sampling with random sampling in the original RBS. The modified RBS produces various training/test sets according to the random seed. Figure 4 shows the pseudocode for the modified RBS [2].

### 2.2. Phase 2: Evaluate the Candidates and Select Best Train/Test Sets

The main goal of Phase 2 is to find the best training/tests from 1000 candidates. We evaluated each candidate according to the workflow shown in Figure 5. Each feature in the dataset was scaled to have a value between 0 and 1, and then histograms were generated for the whole dataset and candidate train/test sets. Based on the histogram data, the similarity in the distribution between the whole dataset and the training set, and between the whole data set and the test set was measured using the Earth Mover's Distance. The final similarity distance for each candidate was obtained by summing the obtained similarity distance for each feature, which reflects the weight relative to the

importance of the feature. Once the similarity distances for all candidates were obtained, we selected the candidate with the smallest distance as the output of the FWS method. We explain the histogram generation, similarity calculation, and feature weighting in the following sections.

```
1   /*
2   Function name: mRBS()
3   Input: DS, k, t
4       (DS: whole dataset, k: number of nearest neighbors, t: ratio of training set)
5   Output: trainset, testSet
6   */
7
8   PDO ← NULL ;              // Array to store degree of overlap (DO) value
9   trainSet ← ∅ ;
10  testSet ← ∅ ;
11
12  FOR i = 1 TO N DO
13  PDO[i] = DO(Pi)          // calculate the DO for each instance
14  END FOR
15
16  FOR i = 0 TO k DO
17  GROUP ← {Pj | Pj ∈ DS and PDO[j] = i }
18  NG ← number of instances in GROUP
19  TR ← NG*t instances of GROUP that are randomly chosen
20  trainSet ← trainSet ∪ TR
21  END FOR
22
23  testSet ← DS − trainSet
24
25  RETURN trainSet, testSet
```

**Figure 4.** Pseudocode of modified R-value-based sampling (RBS).

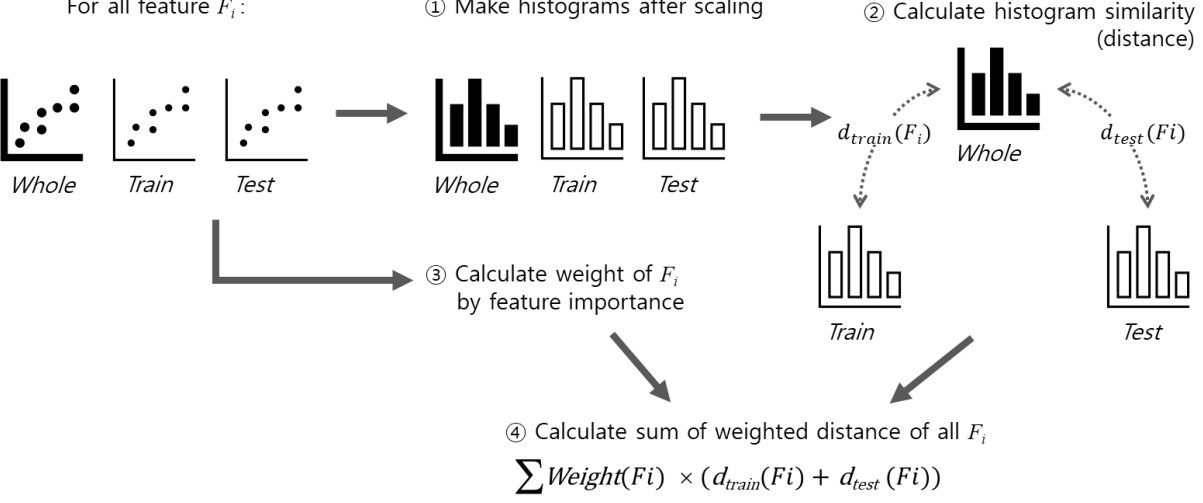

**Figure 5.** The measurement of similarity between original dataset and given train/test sets.

### 2.2.1. Generation of Histograms

The histogram represents the approximated distribution by corresponding a set of real values to an equally wide interval, which is called bin. For example, if a histogram is configured with $n$ bin, it can be defined as a $histogram = \{(bin_i, value_i)|1 \leq i \leq n, where\ bin_k < bin_j\ when\ k < j\}$. The above definition allows the histogram to be represented as a bar for data visualization. However, it is more advantageous to use it as a pure mathematical object containing an approximate data distribution [13,14]. Histograms are mathematical tools that extract compressed characteristic information of a dataset and

play an important role in various fields such as computer vision, image retrieval, and databases [12–15]. We confirmed that the histogram approach is better than the statistical quantile.

This work also views the histogram as a mathematical object and attempts to measure the quantitative similarity between the entire dataset and the candidate dataset. By transforming the real distribution into a histogram, finding train/test sets with the distribution most similar to the entire dataset can be considered the same as the image retrieval problem. Our goal was to find the most similar histogram image of the entire dataset from 1000 candidate histogram images.

### 2.2.2. Measurement of Histogram Similarity

We evaluated the similarity of histograms using distance perspective closeness. Although there are several methods and metrics to obtain similarity distances between histograms [14,15], we exploited the Earth Mover's Distance [12], which adopts a cross-bin scheme. Unlike the bin-by-bin method, cross-bin measurement evaluates not only exactly corresponding bins but also non-responding bins (Figure 6) [12]. It induces less sensitivity to the location of bins and better reflects human-aware similarities [12]. The Earth Mover's Distance is a cross-bin method based on the optimal transport theory, and several studies have demonstrated its superiority [12,15]. In addition, this measurement method has the properties of true distance metrics that satisfy non-negativity, symmetry, and triangle order inequality [15]. In this study, the similarity between datasets is defined as the sum of the histogram distances of all features. Furthermore, the Earth Mover's Distance was calculated using the emdist package in CRAN (https://cran.r-project.org/web/packages/emdist/index.html, accessed on 25 February 2021).

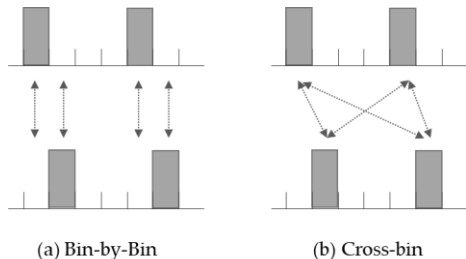

(a) Bin-by-Bin          (b) Cross-bin

**Figure 6.** Comparison of two methods for measurement of histogram similarity.

### 2.2.3. Feature Weighting

Previously, we conceptually defined the similarity between datasets by the distances between features; however, simple distances can be a problem. This is because each feature not only has a different distribution of values, but also has a different degree of contribution to model accuracy. In other words, the same similarity distance between features has different effects on predictive power. For example, although features *A* and *B* have equally strong similarity distances, *A* may have a very strong effect on model accuracy, whereas *B* may have a weak effect. Therefore, when calculating the distances between each feature, we must apply the weight according to the effect of each feature.

There are many methods to evaluate the effects of features, such as information gain and chi-square. We used the Shapley value-based feature importance method [16]. The Shapley value is a method for evaluating the contribution of each feature value in an instance to the model. It takes the idea of game theory to distribute profits fairly according to the contribution of each player. Recently, Covert [16] proposed a method to measure feature importance from a global perspective of the Shapley value as a dataset rather than each instance. This method, called SAGE, has also been published as a Python package. We used this method to obtain feature importance and assign weights when calculating

the similarity distances. The weighted distance between the entire dataset and the given train/test sets was defined as follows:

$$d = \sum_{i=1}^{n} w(f_i) \times (d_{train}(f_i) + d_{test}(f_i)) \tag{1}$$

- $d$: similarity distance of given train/test sets
- $w(f_i)$: weight of $i$-th feature
- $d_{train}(f_i)$: Similarity distance between whole dataset and training set for $i$-th feature
- $d_{test}(f_i)$: Similarity distance between whole dataset and test set for $i$-th feature

The pseudocode for proposed FWS method is described in Figure 7.

```
1    /*
2    Algorithm: FWS
3    Input: DS, k, t, n, bw
4        (DS: whole dataset, k: number of nearest neighbors, t: ratio of training set,
5         n: number of candidates to generate, bw: width of histogram bin)
6    Output: trainSet, testSet
7    */
8
9    PDO ← NULL ;              // Array to store degree of overlap (DO) value
10   FDIST ← NULL ;
11   DIST ← NULL ;
12   NF ← number of features in DS
13   HWH ← Histogram for the whole dataset
14   FI ← array of feature importance
15   testSet ← ∅ ;
16
17   FOR i = 1 TO n DO
18       // generate a candidate
19       CALL mRBS(DS, k, t);
20   CANDIDATE_train[i] ← trainset from mRBS
21   CANDIDATE_test[i] ← testset from mRBS
22
23       // evaluate similarity between DS and given candidate
24   FOR k = 0 TO NF DO
25       HTR ← histogram of k-th feature in trainSet
26       HTS ← histogram of k-th feature in testSet
27       FDIST[k] ← the Earth Mover's Distance between HWH and HTS, HWH and HTR
28   END FOR
29   DIST[i] ← SUM(FDIST* FI)
30   END FOR
31
32   // choose best train/test sets
33   IDX ← index of DIST which has smallest value
34   trainset ← CANDIDATE_train[IDX]
35   testSet ← CANDIDATE_test[IDX]
36   RETURN trainSet, testSet
```

**Figure 7.** The pseudocode of feature-weighted sampling (FWS) method.

### 2.3. Evaluation of FWS Method

To confirm the performance of the proposed sampling method, we compared it with the original RBS, because RBS has traditionally outperformed other methods. Other methods have already been compared with RBS, and we can omit the comparison with other methods. MAI was used as an evaluation metric. For the benchmark test, 20 datasets and 4 classification algorithms were employed.

### 2.3.1. Evaluation Metric: MAI

Measuring the quality of given train/test sets is a difficult issue, because we do not know which ideal train/test sets completely reflect the entire dataset. Kang [2] proposed *MAI* as a solution. He generated 1000 train/test sets by random sampling and measured

the mean accuracies of a classification algorithm. He considered the mean accuracy as the accuracy of ideal train/test sets. In statistics, the mean of large samples converges to the mean of the population. Let us suppose *that AEV* is the mean accuracy from *n* train/test sets. *The AEV* can be defined as follows:

$$AEV = \left(\sum_{i=1}^{n} test\_acc_i \right)/n \qquad (2)$$

where *test_acc$_i$* is the test accuracy generated by the *ith* random sampling. *MAI* is defined by the following equation:

$$MAI = \frac{|ACC - AEV|}{SD} \qquad (3)$$

where *ACC* refers to the test accuracy derived from the classification model for a test set in *n* train/test sets, and *SD* is the standard deviation of test accuracies (*test_acc$_i$*) in *the AEV*. The intuitive meaning of *MAI* is "how far the given *ACC* is from the *AEV*". Therefore, the smaller the *MAI* is, the better. We used the *MAI* as an evaluation metric for the train/test sets.

### 2.3.2. Benchmark Datasets and Classifiers

To compare the proposed FWS and RBS, we used 20 benchmark datasets with various numbers of features (attributes), classes, and instances. The datasets were collected from the UCI Machine Learning Repository (http://archive.ics.uci.edu/mL/, accessed on 25 February 2021) and Kaggle site (https://www.kaggle.com/, accessed on 25 February 2021), and are listed in Table 1. Four classification algorithms, KNN, SVM, RF, and C50 were tested. They are supported by R packages. The packages and parameters used are listed in Table 2. We divided the entire dataset into training and test sets at a 75:25 ratio to build and evaluate the classification models.

**Table 1.** List of benchmark datasets.

| No | Name | # of Features | # of Instances | # of Class |
|----|------|---------------|----------------|------------|
| 1 | audit | 25 | 772 | 2 |
| 2 | avila | 10 | 10,430 | 12 |
| 3 | breastcancer | 30 | 569 | 2 |
| 4 | breastTissue | 9[1] | 106 | 6 |
| 5 | ecoil | 7 | 336 | 8 |
| 6 | Frogs_MFCCs | 22 | 7127 | 3 |
| 7 | gender_classification | 7 | 5001 | 2 |
| 8 | glass | 9 | 214 | 6 |
| 9 | hill_Valley | 100 | 1212 | 2 |
| 10 | ionosphere | 33 | 351 | 2 |
| 11 | iris | 4 | 150 | 3 |
| 12 | liver | 6 | 345 | 2 |
| 13 | music_genre | 26 | 1000 | 10 |
| 14 | pima_diabetes | 8 | 768 | 2 |
| 15 | satimage | 36 | 4435 | 6 |
| 16 | seed | 7 | 210 | 3 |
| 17 | statlog_segment | 16 | 2310 | 7 |
| 18 | wdbc | 30 | 569 | 2 |
| 19 | winequality | 11 | 4893 | 6 |
| 20 | Wireless_Indoor | 7 | 2000 | 4 |

**Table 2.** Summary of classifier and applied parameters.

| Classifier | R Package | Parameter Values |
|---|---|---|
| KNN | class | k = 5 |
| SVM | e1071 | Default |
| RF | randomForest | Default |
| C50 | C50 | trials = 1 |

## 3. Results

In the first phase of generating a candidate train/test set, the *MAI* value was examined while adjusting the *K* value, which determines the sensitivity of category overlap during balanced sampling. We experimented with the influence of *K*. Figure 8 and Table A1 in Appendix A describe the results. The average *MAI* was measured according to *K*. In this experiment, the bin width was fixed at 0.2. As we can see, the overall performance was the best when *K* was 3. When the value of *K* increased, the number of groups also increased, and instances in a specific group tended to become sparse. When the instances of each group were insufficient, the diversity of the distribution could not be secured. Therefore, a small number of *K* is advantageous for the proposed method.

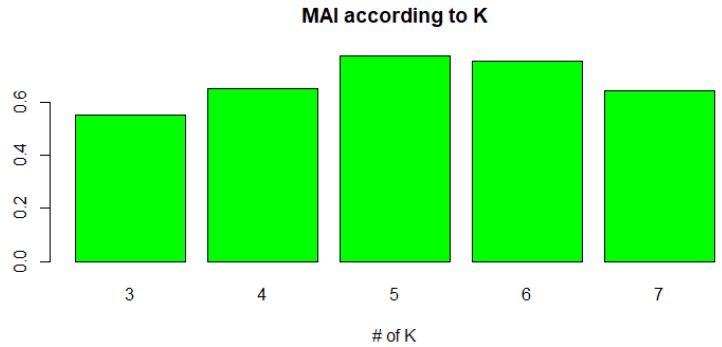

**Figure 8.** Mean accuracy index (*MAI*) of FWS according to K (bin width = 0.2).

The value of bin width is another important parameter for the proposed FWS. Therefore, we experimented with the influence of the bin width. We tested the value 0.2 (the number of bins is 5), 0.1 (the number of bins is 10), and 0.05 (the number of bins is 20), and *K* was fixed at 3. Figure 9 and Table A2 in Appendix A summarize the results. When the bin width was 0.2, the performance was slightly good, but there was no significant difference overall. In another experiment, we confirmed that 0.2 was the best for multi-class datasets (number of classes > 2), whereas 0.05 was the best for binary-class datasets. Therefore, we used 0.05 and 0.2 as hybrid methods for the final FWS method.

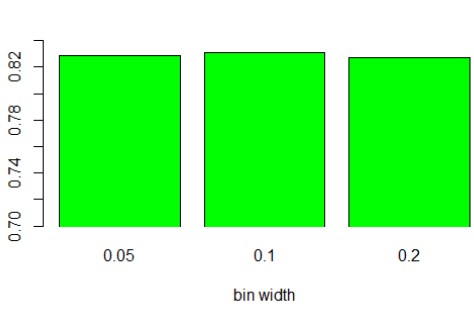

**Figure 9.** *MAI* of FWS according to bin width (*K* = 3).

Table 3 shows the final experimental results when k = 3 and the bin width is hybrid. The 61 cases (76%) of *MAI* of FWS were better than those of RBS, and RBS was better than FWS in 19 cases (24%). This result indicates that FWS is a more improved method than original RBS and previous methods. The details are discussed in the next section.

**Table 3.** Final experimental results.

| Dataset | Classifier | MA | SD | RBS Accuracy | FWS Accuracy | RBS *MAI* | FWS *MAI* |
|---|---|---|---|---|---|---|---|
| 1 | C50 | 0.998 | 0.004 | 0.99479 | 0.990 | **0.972** | 2.383 |
| | KNN | 0.957 | 0.014 | 0.94792 | 0.949 | 0.628 | **0.552** |
| | RF | 0.998 | 0.003 | 0.99479 | 1.000 | 1.083 | **0.479** |
| | SVM | 0.969 | 0.012 | 0.9375 | 0.959 | 2.623 | **0.792** |
| 2 | C50 | 0.974 | 0.004 | 0.97694 | 0.974 | 0.645 | **0.076** |
| | KNN | 0.698 | 0.007 | 0.69254 | 0.702 | 0.758 | **0.474** |
| | RF | 0.979 | 0.003 | 0.97771 | 0.975 | **0.388** | 1.233 |
| | SVM | 0.69 | 0.011 | 0.68255 | 0.678 | **0.652** | 1.075 |
| 3 | C50 | 0.936 | 0.021 | 0.986 | 0.951 | 2.389 | **0.741** |
| | KNN | 0.968 | 0.013 | 0.986 | 0.965 | 1.348 | **0.202** |
| | RF | 0.959 | 0.017 | 0.972 | 0.958 | 0.740 | **0.061** |
| | SVM | 0.974 | 0.012 | 0.993 | 0.972 | 1.532 | **0.155** |
| 4 | C50 | 0.665 | 0.069 | 0.708 | 0.676 | 0.632 | **0.161** |
| | KNN | 0.661 | 0.078 | 0.625 | 0.595 | **0.458** | 0.845 |
| | RF | 0.699 | 0.066 | 0.583 | 0.649 | 1.746 | **0.760** |
| | SVM | 0.598 | 0.070 | 0.583 | 0.514 | **0.215** | 1.213 |
| 5 | C50 | 0.803 | 0.036 | 0.783 | 0.822 | 0.546 | **0.538** |
| | KNN | 0.850 | 0.027 | 0.855 | 0.856 | **0.209** | 0.214 |
| | RF | 0.860 | 0.029 | 0.831 | 0.878 | 0.987 | **0.618** |
| | SVM | 0.807 | 0.061 | 0.735 | 0.778 | 1.191 | **0.485** |
| 6 | C50 | 0.963 | 0.005 | 0.967 | 0.961 | 0.754 | **0.268** |
| | KNN | 0.992 | 0.002 | 0.991 | 0.992 | 0.319 | **0.243** |
| | RF | 0.987 | 0.003 | 0.984 | 0.987 | 0.973 | **0.159** |
| | SVM | 0.992 | 0.002 | 0.990 | 0.990 | 1.037 | **0.737** |
| 7 | C50 | 0.972 | 0.004 | 0.970 | 0.970 | 0.369 | **0.347** |
| | KNN | 0.965 | 0.005 | 0.970 | 0.965 | 1.086 | **0.122** |
| | RF | 0.974 | 0.004 | 0.977 | 0.973 | 0.655 | **0.317** |
| | SVM | 0.972 | 0.004 | 0.973 | 0.970 | 0.298 | **0.252** |
| 8 | C50 | 0.686 | 0.055 | 0.615 | 0.694 | 1.275 | **0.145** |
| | KNN | 0.634 | 0.049 | 0.596 | 0.645 | 0.768 | **0.231** |
| | RF | 0.779 | 0.048 | 0.750 | 0.758 | 0.608 | **0.439** |
| | SVM | 0.686 | 0.048 | 0.673 | 0.629 | **0.276** | 1.185 |
| 9 | C50 | 0.505 | 0.002 | 0.505 | 0.503 | **0.110** | 0.807 |
| | KNN | 0.548 | 0.025 | 0.558 | 0.549 | 0.380 | **0.025** |
| | RF | 0.600 | 0.026 | 0.653 | 0.601 | 2.061 | **0.065** |
| | SVM | 0.515 | 0.017 | 0.545 | 0.510 | 1.776 | **0.297** |
| 10 | C50 | 0.9 | 0.03 | 0.862 | 0.890 | 1.276 | **0.344** |
| | KNN | 0.844 | 0.029 | 0.839 | 0.846 | 0.169 | **0.071** |
| | RF | 0.934 | 0.022 | 0.954 | 0.923 | 0.882 | **0.500** |
| | SVM | 0.942 | 0.022 | 0.943 | 0.923 | **0.017** | 0.848 |
| 11 | C50 | 0.938 | 0.036 | 0.944 | 0.951 | **0.174** | 0.365 |
| | KNN | 0.96 | 0.031 | 0.944 | 0.951 | 0.484 | **0.267** |
| | RF | 0.956 | 0.03 | 0.944 | 0.951 | 0.376 | **0.152** |
| | SVM | 0.961 | 0.031 | 0.944 | 0.951 | 0.538 | **0.316** |

**Table 3.** *Cont.*

| Dataset | Classifier | MA | SD | RBS Accuracy | FWS Accuracy | RBS *MAI* | FWS *MAI* |
|---|---|---|---|---|---|---|---|
| 12 | C50 | 0.648 | 0.048 | 0.756 | 0.637 | 2.252 | **0.213** |
|  | KNN | 0.607 | 0.045 | 0.605 | 0.560 | **0.062** | 1.047 |
|  | RF | 0.725 | 0.043 | 0.767 | 0.714 | 0.983 | **0.257** |
|  | SVM | 0.693 | 0.04 | 0.721 | 0.648 | **0.689** | 1.112 |
| 13 | C50 | 0.485 | 0.03 | 0.484 | 0.466 | **0.029** | 0.641 |
|  | KNN | 0.616 | 0.027 | 0.568 | 0.617 | 1.790 | **0.058** |
|  | RF | 0.645 | 0.027 | 0.628 | 0.652 | 0.610 | **0.256** |
|  | SVM | 0.654 | 0.028 | 0.648 | 0.633 | **0.229** | 0.784 |
| 14 | C50 | 0.736 | 0.029 | 0.724 | 0.745 | 0.423 | **0.289** |
|  | KNN | 0.734 | 0.026 | 0.719 | 0.724 | 0.570 | **0.349** |
|  | RF | 0.762 | 0.025 | 0.734 | 0.760 | 1.109 | **0.078** |
|  | SVM | 0.761 | 0.026 | 0.724 | 0.755 | 1.405 | **0.209** |
| 15 | C50 | 0.857 | 0.010 | 0.859 | 0.852 | **0.225** | 0.454 |
|  | KNN | 0.901 | 0.008 | 0.898 | 0.898 | 0.343 | **0.318** |
|  | RF | 0.910 | 0.008 | 0.914 | 0.915 | **0.548** | 0.667 |
|  | SVM | 0.891 | 0.008 | 0.892 | 0.894 | **0.056** | 0.425 |
| 16 | C50 | 0.908 | 0.039 | 0.882 | 0.912 | 0.664 | **0.100** |
|  | KNN | 0.928 | 0.032 | 0.882 | 0.930 | 1.421 | **0.066** |
|  | RF | 0.927 | 0.035 | 0.882 | 0.930 | 1.276 | **0.085** |
|  | SVM | 0.929 | 0.030 | 0.882 | 0.930 | 1.533 | **0.023** |
| 17 | C50 | 0.964 | 0.008 | 0.977 | 0.973 | 1.644 | **1.062** |
|  | KNN | 0.959 | 0.007 | 0.963 | 0.956 | 0.661 | **0.424** |
|  | RF | 0.978 | 0.006 | 0.980 | 0.974 | **0.465** | 0.613 |
|  | SVM | 0.944 | 0.008 | 0.955 | 0.950 | 1.340 | **0.826** |
| 18 | C50 | 0.936 | 0.021 | 0.986 | 0.951 | 0.895 | **0.741** |
|  | KNN | 0.968 | 0.013 | 0.986 | 0.965 | 0.965 | **0.202** |
|  | RF | 0.959 | 0.017 | 0.972 | 0.958 | 0.951 | **0.061** |
|  | SVM | 0.974 | 0.012 | 0.993 | 0.972 | 0.979 | **0.155** |
| 19 | C50 | 0.573 | 0.014 | 0.636 | 0.581 | 4.636 | **0.636** |
|  | KNN | 0.543 | 0.012 | 0.571 | 0.541 | 2.329 | **0.099** |
|  | RF | 0.684 | 0.011 | 0.723 | 0.699 | 3.404 | **1.309** |
|  | SVM | 0.571 | 0.011 | 0.577 | 0.572 | 0.497 | **0.033** |
| 20 | C50 | 0.97 | 0.007 | 0.972 | 0.968 | 0.296 | **0.226** |
|  | KNN | 0.984 | 0.005 | 0.98 | 0.980 | 0.732 | **0.693** |
|  | RF | 0.984 | 0.005 | 0.978 | 0.984 | 1.040 | **0.086** |
|  | SVM | 0.981 | 0.005 | 0.98 | 0.984 | **0.158** | 0.610 |

MA: mean classification accuracy, SD: standard deviation.

## 4. Discussion

RBS is an efficient sampling method compared to the previous methods. The proposed FWS is a more improved method than RBS. Figure 10 shows how much more advanced FWS is than RBS. As shown in Figure 10a, the average *MAI* value of FWS was 0.460, whereas that of RBS was 0.920. As we can see, the smaller the *MAI*, the better. Therefore, FWS improved MAI by 56% compared to RBS. Figure 10b compares the standard deviations of *the MAI*. The standard deviation of FWS was 0.403, whereas that of RBS was 0.779. In other words, this means the distribution of the *MAI* value of the FWS was smaller than that of the RBS. FWS yielded more stable sampling results than RBS. Figure 10c shows the range of *MAI*. The range was calculated as (maximum of *MAI*) − (minimum of *MAI*). It also shows the distribution of *MAI* values. The ranges of FWS and RBS were 2.359 and 4.619, respectively. The fluctuation range of the FWS was smaller than that of the RBS. All statistics in Figure 10 show that FWS is a more stable and accurate method than RBS.

Furthermore, it is proven that the similarity of distribution between the train/test sets and whole dataset is an important factor for their ideal splitting.

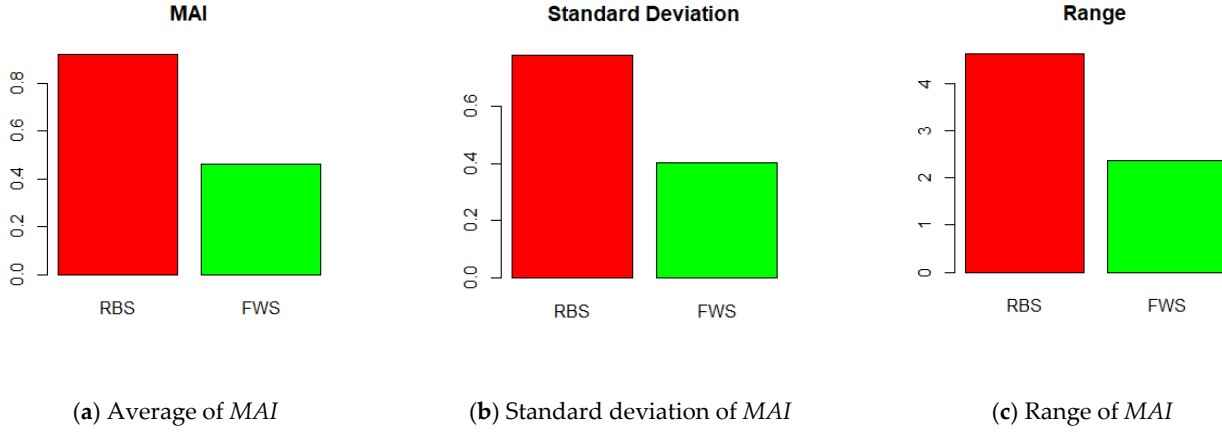

(**a**) Average of *MAI*                    (**b**) Standard deviation of *MAI*                    (**c**) Range of *MAI*

**Figure 10.** Comparison of RBS and FWS.

In the development of the prediction model, the quality of the features determines the performance of the model. In general, the influence of features is greater than that of classification algorithms [17]. Therefore, considering the feature weight for distance calculation in the classification is reasonable. Figure 11 shows the influence of feature weighting in the FWS method. We compared FWS with and without feature weights. In the average of *MAI*, the "without case" was 0.634 whereas "with case" was 0.490 (Figure 11a). This means that feature weighting improved the performance of the FWS. In terms of the standard deviation, both cases were similar (Figure 11b). The ranges of "with case" and "without case" were 2.314 and 1.801, respectively (Figure 11c). This is because the maximum value of "with case" was large, which is less important than the standard deviation.

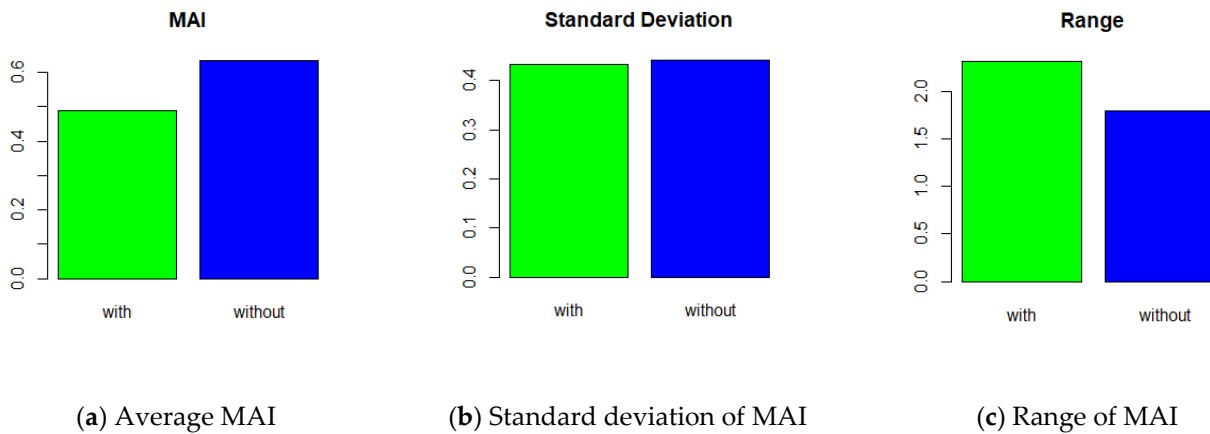

(**a**) Average MAI                    (**b**) Standard deviation of MAI                    (**c**) Range of MAI

**Figure 11.** FWS with and without feature-weighted distance.

We analyzed the variance in the *MAI* according to the number of classes. In the result of RBS, the *MAI* value of binary-class datasets was higher than that of multi-class datasets, whereas the difference was not large in FWS (Figure 12). This means that FWS is not influenced by the variance in the class number. FWS is a more stable method than RBS.

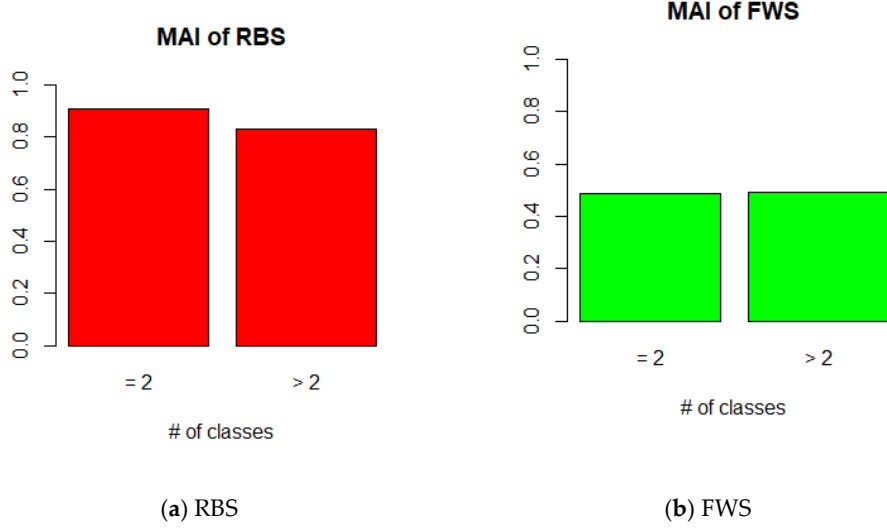

**(a)** RBS           **(b)** FWS

**Figure 12.** *MAI* of FWS according to number of class (K = 3, bin width = hybrid).

In this study, we confirmed that the similarity of distribution between the original dataset and train/test sets is an important factor for accurate sampling. Furthermore, feature-weighted distance calculation can improve the sampling performance. If we use the proposed FWS for splitting train/test sets, we can more accurately evaluate the classification models. In our experiment, FWS performed better than RBS in 61 of 80 cases of train/test sets. This shows that the FWS has room for further improvement and is a topic for further research.

**Author Contributions:** Conceptualization, S.O.; methodology, H.S. and S.O.; software, H.S.; validation, H.S. and S.O.; formal analysis, H.S. and S.O.; investigation, H.S. and S.O.; resources, H.S.; data curation, H.S.; writing—original draft preparation, H.S. and S.O.; writing—review and editing, H.S. and S.O.; visualization, H.S. and S.O.; supervision, S.O.; project administration, S.O.; funding acquisition, S.O. All authors have read and agreed to the published version of the manuscript.

**Funding:** The present research was supported by the research fund of Dankook University in 2019.

**Institutional Review Board Statement:** Not applicable.

**Informed Consent Statement:** Not applicable.

**Conflicts of Interest:** The authors declare no conflict of interest.

## Appendix A

**Table A1.** MAI of FWS according to K (bin width = 0.2).

| No | Classifier | K = 3 | K = 4 | K = 5 | K = 6 | K = 7 |
|----|-----------|-------|-------|-------|-------|-------|
| 1 | C50 | 2.383 | 2.368 | 0.499 | 0.499 | 0.499 |
|   | KNN | 0.552 | 0.534 | 1.252 | 0.480 | 0.480 |
|   | RF | 0.479 | 0.479 | 0.479 | 1.020 | 0.479 |
|   | SVM | 0.792 | 0.775 | 0.940 | 1.568 | 0.543 |
| 2 | C50 | 0.076 | 0.491 | 0.232 | 0.719 | 0.120 |
|   | KNN | 0.474 | 0.241 | 0.515 | 2.064 | 0.390 |
|   | RF | 1.233 | 0.211 | 1.328 | 0.153 | 3.192 |
|   | SVM | 1.075 | 1.557 | 0.058 | 1.175 | 0.418 |

**Table A1.** *Cont.*

| No | Classifier | K = 3 | K = 4 | K = 5 | K = 6 | K = 7 |
|----|-----------|-------|-------|-------|-------|-------|
| 3 | C50 | 0.078 | 0.446 | 1.189 | 0.835 | 0.164 |
| | KNN | 1.766 | 0.166 | 1.195 | 0.659 | 0.639 |
| | RF | 0.351 | 0.027 | 0.379 | 0.414 | 0.407 |
| | SVM | 0.721 | 0.682 | 0.682 | 0.663 | 0.644 |
| 4 | C50 | 0.161 | 1.010 | 0.932 | 1.293 | 0.079 |
| | KNN | 0.845 | 0.532 | 1.135 | 0.458 | 0.313 |
| | RF | 0.760 | 0.055 | 1.117 | 1.495 | 1.112 |
| | SVM | 1.213 | 0.332 | 0.023 | 1.407 | 0.107 |
| 5 | C50 | 0.538 | 0.078 | 0.414 | 0.322 | 1.122 |
| | KNN | 0.214 | 0.286 | 0.326 | 0.165 | 0.430 |
| | RF | 0.618 | 0.251 | 1.441 | 0.502 | 1.316 |
| | SVM | 0.485 | 0.576 | 0.182 | 0.730 | 0.731 |
| 6 | C50 | 0.268 | 0.837 | 0.832 | 1.184 | 1.943 |
| | KNN | 0.243 | 0.027 | 0.025 | 0.576 | 0.845 |
| | RF | 0.159 | 0.503 | 0.167 | 0.948 | 0.274 |
| | SVM | 0.737 | 0.447 | 0.729 | 0.447 | 0.727 |
| 7 | C50 | 0.347 | 0.524 | 0.337 | 0.883 | 0.160 |
| | KNN | 0.122 | 0.466 | 0.241 | 0.066 | 0.235 |
| | RF | 0.317 | 0.905 | 0.484 | 1.097 | 0.084 |
| | SVM | 0.252 | 1.006 | 0.810 | 0.620 | 0.246 |
| 8 | C50 | 0.145 | 0.920 | 1.169 | 2.073 | 0.344 |
| | KNN | 0.231 | 0.346 | 0.361 | 0.689 | 1.287 |
| | RF | 0.439 | 1.023 | 0.502 | 0.140 | 0.502 |
| | SVM | 1.185 | 2.047 | 0.529 | 1.490 | 0.217 |
| 9 | C50 | 0.807 | 0.075 | 7.890 | 1.820 | 0.818 |
| | KNN | 1.039 | 0.482 | 0.517 | 0.719 | 0.144 |
| | RF | 0.690 | 0.633 | 0.830 | 0.778 | 0.085 |
| | SVM | 1.652 | 2.145 | 0.086 | 1.531 | 0.086 |
| 10 | C50 | 0.020 | 0.448 | 0.508 | 0.832 | 1.027 |
| | KNN | 0.675 | 0.183 | 0.781 | 0.123 | 0.241 |
| | RF | 0.500 | 0.054 | 0.115 | 1.340 | 0.948 |
| | SVM | 0.359 | 1.253 | 0.232 | 1.212 | 1.123 |
| 11 | C50 | 0.365 | 0.398 | 0.273 | 0.429 | 0.273 |
| | KNN | 0.267 | 0.230 | 0.533 | 0.195 | 0.230 |
| | RF | 0.152 | 0.114 | 0.114 | 0.844 | 0.114 |
| | SVM | 0.316 | 0.278 | 0.500 | 0.242 | 0.500 |
| 12 | C50 | 0.930 | 0.702 | 1.616 | 0.899 | 1.128 |
| | KNN | 0.312 | 0.912 | 0.912 | 0.422 | 0.067 |
| | RF | 0.514 | 0.768 | 0.512 | 1.283 | 0.512 |
| | SVM | 0.839 | 1.615 | 2.160 | 0.294 | 1.112 |
| 13 | C50 | 0.641 | 0.311 | 0.825 | 0.634 | 1.249 |
| | KNN | 0.058 | 0.498 | 1.366 | 0.460 | 0.133 |
| | RF | 0.256 | 1.089 | 1.402 | 0.063 | 0.787 |
| | SVM | 0.784 | 1.250 | 0.619 | 0.414 | 1.642 |
| 14 | C50 | 0.462 | 0.185 | 1.220 | 0.720 | 0.605 |
| | KNN | 1.024 | 0.295 | 0.100 | 0.630 | 1.349 |
| | RF | 0.328 | 0.375 | 0.840 | 0.018 | 0.135 |
| | SVM | 0.405 | 0.619 | 0.746 | 0.502 | 0.453 |
| 15 | C50 | 0.454 | 0.201 | 0.585 | 0.280 | 0.042 |
| | KNN | 0.318 | 0.149 | 0.161 | 0.918 | 0.537 |
| | RF | 0.667 | 0.091 | 0.129 | 1.118 | 1.073 |
| | SVM | 0.425 | 0.150 | 0.643 | 0.506 | 0.678 |

**Table A1.** *Cont.*

| No | Classifier | K = 3 | K = 4 | K = 5 | K = 6 | K = 7 |
|----|-----------|-------|-------|-------|-------|-------|
| 16 | C50 | 0.100 | 0.138 | 0.176 | 0.138 | 0.247 |
|    | KNN | 0.066 | 1.184 | 0.391 | 0.976 | 0.723 |
|    | RF  | 0.085 | 0.614 | 0.819 | 1.364 | 0.217 |
|    | SVM | 0.023 | 0.063 | 0.101 | 0.063 | 0.174 |
| 17 | C50 | 1.062 | 2.067 | 0.677 | 0.145 | 2.023 |
|    | KNN | 0.424 | 0.382 | 1.763 | 0.807 | 1.272 |
|    | RF  | 0.613 | 1.997 | 0.570 | 1.117 | 1.959 |
|    | SVM | 0.826 | 1.581 | 1.552 | 0.512 | 1.727 |
| 18 | C50 | 0.078 | 0.446 | 1.189 | 0.835 | 0.164 |
|    | KNN | 1.766 | 0.166 | 1.195 | 0.659 | 0.639 |
|    | RF  | 0.351 | 0.027 | 0.379 | 0.414 | 0.407 |
|    | SVM | 0.721 | 0.682 | 0.682 | 0.663 | 0.644 |
| 19 | C50 | 0.636 | 0.062 | 0.347 | 1.515 | 0.388 |
|    | KNN | 0.099 | 2.152 | 0.724 | 0.747 | 0.243 |
|    | RF  | 1.309 | 0.751 | 2.264 | 0.566 | 0.302 |
|    | SVM | 0.033 | 1.142 | 0.343 | 1.126 | 2.135 |
| 20 | C50 | 0.226 | 0.459 | 0.366 | 0.737 | 0.381 |
|    | KNN | 0.693 | 0.656 | 0.104 | 0.110 | 0.266 |
|    | RF  | 0.086 | 0.961 | 0.827 | 1.319 | 0.120 |
|    | SVM | 0.610 | 1.173 | 0.819 | 0.448 | 0.440 |
|    | **mean** | **0.554** | **0.654** | **0.775** | **0.754** | **0.645** |

**Table A2.** MAI of FWS according to bin width (K = 3).

| No | Classifier | bw = 0.05 | bw = 0.1 | bw = 0.2 |
|----|-----------|-----------|----------|----------|
| 1 | C50 | 2.383 | 2.383 | 2.383 |
|   | KNN | 0.552 | 0.552 | 0.552 |
|   | RF  | 0.479 | 0.479 | 0.479 |
|   | SVM | 0.792 | 0.792 | 0.792 |
| 2 | C50 | 0.076 | 1.519 | 1.519 |
|   | KNN | 0.474 | 0.615 | 0.615 |
|   | RF  | 1.233 | 1.016 | 1.016 |
|   | SVM | 1.075 | 1.040 | 1.040 |
| 3 | C50 | 0.741 | 0.078 | 0.078 |
|   | KNN | 0.202 | 1.766 | 1.766 |
|   | RF  | 0.061 | 0.351 | 0.351 |
|   | SVM | 0.155 | 0.721 | 0.721 |
| 4 | C50 | 0.161 | 0.941 | 0.161 |
|   | KNN | 0.188 | 0.188 | 0.845 |
|   | RF  | 0.871 | 1.687 | 0.760 |
|   | SVM | 0.054 | 0.719 | 1.213 |
| 5 | C50 | 0.538 | 0.538 | 0.538 |
|   | KNN | 0.214 | 0.214 | 0.214 |
|   | RF  | 0.618 | 0.618 | 0.618 |
|   | SVM | 0.485 | 0.485 | 0.485 |
| 6 | C50 | 0.268 | 0.268 | 0.268 |
|   | KNN | 0.243 | 0.243 | 0.243 |
|   | RF  | 0.159 | 0.159 | 0.159 |
|   | SVM | 0.737 | 0.737 | 0.737 |

**Table A2.** *Cont.*

| No | Classifier | bw = 0.05 | bw = 0.1 | bw = 0.2 |
|---|---|---|---|---|
| 7 | C50 | 0.347 | 0.347 | 0.347 |
|   | KNN | 0.122 | 0.122 | 0.122 |
|   | RF | 0.317 | 0.317 | 0.317 |
|   | SVM | 0.252 | 0.252 | 0.252 |
| 8 | C50 | 0.148 | 0.148 | 0.145 |
|   | KNN | 0.888 | 0.888 | 0.231 |
|   | RF | 0.236 | 0.236 | 0.439 |
|   | SVM | 0.479 | 0.479 | 1.185 |
| 9 | C50 | 0.807 | 0.807 | 0.807 |
|   | KNN | 0.025 | 0.640 | 1.039 |
|   | RF | 0.065 | 0.060 | 0.690 |
|   | SVM | 0.297 | 1.662 | 1.652 |
| 10 | C50 | 0.344 | 0.344 | 0.020 |
|   | KNN | 0.071 | 0.071 | 0.675 |
|   | RF | 0.500 | 0.500 | 0.500 |
|   | SVM | 0.848 | 0.848 | 0.359 |
| 11 | C50 | 0.365 | 0.365 | 0.365 |
|   | KNN | 0.514 | 0.514 | 0.267 |
|   | RF | 0.152 | 0.152 | 0.152 |
|   | SVM | 0.316 | 0.316 | 0.316 |
| 12 | C50 | 0.213 | 0.930 | 0.930 |
|   | KNN | 1.047 | 0.312 | 0.312 |
|   | RF | 0.257 | 0.514 | 0.514 |
|   | SVM | 1.112 | 0.839 | 0.839 |
| 13 | C50 | 1.667 | 1.667 | 0.641 |
|   | KNN | 0.483 | 0.483 | 0.058 |
|   | RF | 1.975 | 1.975 | 0.256 |
|   | SVM | 1.056 | 1.056 | 0.784 |
| 14 | C50 | 0.289 | 0.462 | 0.462 |
|   | KNN | 0.349 | 1.024 | 1.024 |
|   | RF | 0.078 | 0.328 | 0.328 |
|   | SVM | 0.209 | 0.405 | 0.405 |
| 15 | C50 | 1.290 | 0.197 | 0.454 |
|   | KNN | 0.206 | 0.802 | 0.318 |
|   | RF | 0.839 | 0.839 | 0.667 |
|   | SVM | 0.134 | 0.134 | 0.425 |
| 16 | C50 | 0.348 | 0.348 | 0.100 |
|   | KNN | 1.033 | 1.033 | 0.066 |
|   | RF | 0.418 | 0.418 | 0.085 |
|   | SVM | 1.127 | 1.127 | 0.023 |
| 17 | C50 | 1.697 | 0.419 | 1.062 |
|   | KNN | 1.368 | 0.896 | 0.424 |
|   | RF | 0.240 | 1.467 | 0.613 |
|   | SVM | 1.843 | 0.210 | 0.826 |
| 18 | C50 | 0.741 | 0.078 | 0.078 |
|   | KNN | 0.202 | 1.766 | 1.766 |
|   | RF | 0.061 | 0.351 | 0.351 |
|   | SVM | 0.155 | 0.721 | 0.721 |
| 19 | C50 | 0.695 | 0.636 | 0.636 |
|   | KNN | 0.842 | 0.099 | 0.099 |
|   | RF | 0.955 | 1.309 | 1.309 |
|   | SVM | 0.783 | 0.033 | 0.033 |

**Table A2.** *Cont.*

| No | Classifier | bw = 0.05 | bw = 0.1 | bw = 0.2 |
|----|-----------|-----------|----------|----------|
| 20 | C50 | 0.226 | 0.055 | 0.226 |
| | KNN | 0.693 | 0.080 | 0.693 |
| | RF | 0.448 | 1.362 | 0.086 |
| | SVM | 0.610 | 0.244 | 0.610 |
| | **mean** | **0.827** | **0.831** | **0.829** |

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
