# Peer review of "Feature-Weighted Sampling for Proper Evaluation of Classification Models"

_applsci, doi:10.3390/app11052039_

Round 1
Reviewer 1 Report
I only have a couple of minor things to comment:
- Line 203: triangle order quality. Do you mean equality?
- Line 362: it states "In our experiment, RBS performed better than FWS in 61 of 80 cases of train/test sets". If this an error and the authors swapped RBS and FWS? If not, please explain futher.
Author Response
(1) Line 203: triangle order quality. Do you mean equality?
Answer: ‘triangle order inequality’ is correct. We fix it.
(2) Line 362: it states "In our experiment, RBS performed better than FWS in 61 of 80 cases of train/test sets". If this an error and the authors swapped RBS and FWS? If not, please explain further.
Answer: It is a misprint. FWS is better than RBS. We fix it.
Reviewer 2 Report
The manuscript presents a new sampling method for the accurate evaluation of a classification model in order to avoid the problem of evaluating the performance of a classifier with “an easy-to-classify” test set. The paper is well-written and the proposed research is a key topic when it comes to the evaluation of machine learning models.
However, there are some issues in the manuscript that must be addressed.
- It should be more deeply discussed why k-fold-cross validation and holdouts methods are out of the scope of this research work. Why are they different?
- It is not clear to me how you have performed the feature weighting. Reading line 227, I imagine it is done using the whole dataset (which includes all the possible test sets). Would not this give information about which could be the feature weights of the test sets? How is this different from performing parameter tuning using the whole dataset instead of only the train set?
- Have you tried to use other methods for distance computation than Earth mover’s distance? If so you should mention it in the manuscript.
- Also, Instead of using histograms, have you considered using other statistical values for dataset representation? If so you should mention it in the manuscript.
Author Response
(1) It should be more deeply discussed why k-fold-cross validation and holdouts methods are out of the scope of this research work. Why are they different?
Answer: We add the sentence as follows:
Both k-fold cross-validation and holdouts methods produce multiple train/test sets, and as a result, they make multiple prediction models. We cannot know which is a desirable model.
(2) It is not clear to me how you have performed the feature weighting. Reading line 227, I imagine it is done using the whole dataset (which includes all the possible test sets). Would not this give information about which could be the feature weights of the test sets? How is this different from performing parameter tuning using the whole dataset instead of only the train set?
Answer: We also know that information of test set should not be used to the process of model building. As you can see, proposed method is not about model building. The goal of proposed method is to find a train/test sets that has equal distribution of whole dataset. Therefore, using the information of whole dataset is no problem. In our experiment, feature weights are global constant values; they are independent from specific model. Therefore, our feature weights commonly applied on all candidate train/test sets. No candidate gains more profit or loss from the feature weights.
(3) Have you tried to use other methods for distance computation than Earth mover’s distance? If so you should mention it in the manuscript.
Answer: We revise the related sentence as follows:
To calculate the distance between the original dataset and train/test sets, we tested bhattacharyya distance [10], histogram Intersection [11], and Earth mover’s distance [12]. Finally, we adopted the Earth Mover’s Distance.
(4) Also, instead of using histograms, have you considered using other statistical values for dataset representation? If so you should mention it in the manuscript.
Answer: We revise the related sentence as follows:
We confirmed that the histogram approach is better than the statistical quantile.
Round 2
Reviewer 2 Report
I agree with the performed changes.